# Patient-Specific Surgical Correction of Adolescent Idiopathic Scoliosis: A Systematic Review

**DOI:** 10.3390/children11010106

**Published:** 2024-01-15

**Authors:** Federico Solla, Brice Ilharreborde, Jean-Luc Clément, Emma O. Rose, Marco Monticone, Carlo M. Bertoncelli, Virginie Rampal

**Affiliations:** 1Paediatric Orthopaedic Unit, Lenval Foundation, 57, Avenue de la Californie, 06200 Nice, France; clement.jluc@wanadoo.fr (J.-L.C.); carlo.bertoncelli@hpu.lenval.com (C.M.B.); virginie.rocher-rampal@hpu.lenval.com (V.R.); 2Paediatric Orthopaedic Unit, Hôpital Robert Debré, AP-HP, 75019 Paris, France; brice.ilharreborde@aphp.fr; 3Krieger School of Arts & Sciences, Homewood Campus, John Hopkins University, Baltimore, MD 21218, USA; 4Department of Surgical Sciences, University of Cagliari, 09124 Cagliari, Italy; marco.monticone@unica.it

**Keywords:** children, thoracic spine, rods, planning, thoracic kyphosis, pre-bent, contouring

## Abstract

The restoration of sagittal alignment is fundamental to the surgical correction of adolescent idiopathic scoliosis (AIS). Despite established techniques, some patients present with inadequate postoperative thoracic kyphosis (TK), which may increase the risk of proximal junctional kyphosis (PJK) and imbalance. There is a lack of knowledge concerning the effectiveness of patient-specific rods (PSR) with measured sagittal curves in achieving a TK similar to that planned in AIS surgery, the factors influencing this congruence, and the incidence of PJK after PSR use. This is a systematic review of all types of studies reporting on the PSR surgical correction of AIS, including research articles, proceedings, and gray literature between 2013 and December 2023. From the 28,459 titles identified in the literature search, 81 were assessed for full-text reading, and 7 studies were selected. These included six cohort studies and a comparative study versus standard rods, six monocentric and one multicentric, three prospective and four retrospective studies, all with a scientific evidence level of 4 or 3. They reported a combined total of 355 AIS patients treated with PSR. The minimum follow-up was between 4 and 24 months. These studies all reported a good match between predicted and achieved TK, with the main difference ranging from 0 to 5 degrees, *p* > 0.05, despite the variability in surgical techniques and the rods’ properties. There was no proximal junctional kyphosis, whereas the current rate from the literature is between 15 and 46% with standard rods. There are no specific complications related to PSR. The exact role of the type of implants is still unknown. The preliminary results are, therefore, encouraging and support the use of PSR in AIS surgery.

## 1. Introduction

The pathological coronal curve of adolescent idiopathic scoliosis (AIS) combined with the sagittal alignment causes a 3D deformity [1,2,3]. This automatically leads to a modification of the sagittal curvatures, which manifests, in most cases, in a flat back with proximal lumbar hypolordosis, thoracic hypokyphosis, and cervical hypolordosis or kyphosis [4,5,6].

Nevertheless, most patients with “unfused” AIS remain balanced on the sagittal plane thanks to the spine’s flexibility, which allows for spontaneous equilibration [7,8].

Current correction techniques using high-density anchors allow for a relevant reduction in the coronal deformity (from 65 to 80%) [8]. In addition to coronal outcomes, sagittal results strongly affect long-term quality of life [9,10,11] and the degeneration of uninstrumented levels for both the cervical [12,13,14,15] and lumbar [16].

The majority of AIS procedures include thoracic spine fixation, either for the main curve (Lenke 1–4) or for the thoracic counter-curve of Lenke 6 [8,17]. Consequently, the deformity correction requires an appropriate instrumented thoracic spine alignment.

Moreover, many publications reporting AIS postoperative outcomes have emphasized the risk of thoracic hypokyphosis after posterior fusion [8,11,18,19]. Therefore, several authors have taken an interest in this problem, underlining the need to obtain a “normal” postoperative thoracic kyphosis (TK) [20,21,22]. The normal TK is currently accepted to be between 10° and 40° (according to Lenke’s classification) or between 20° and 50° [22,23,24]. However, recent works have suggested that there should not be the same normal TK for all individuals but rather a patient-specific TK adapted to the individual lumbo-pelvic parameters [24,25,26]. Therefore, the targeted TK for each patient remains debatable, as well as its distribution (i.e., the number of vertebrae in TK) and the location of the TK apex (TKA) [22,25]. In addition, the way of measuring TK is not unanimous, with various methods being used, based on either predefined anatomical landmarks (e.g., T4–T12) or functional ones (e.g., global TK) 2,20,22].

Moreover, the insufficient restoration of TK increases the risk of proximal junctional kyphosis (PJK) or proximal junctional failure [27,28]. Indeed, PJK allows for the patient to regain their sagittal balance by accentuating the kyphosis above the fusion [26,29]. These iatrogenic PJKs come up frequently in the literature, involving up to 46% of patients, and can usually be detected early (within 4 months postoperative) [13,28]. Even if few of them require revision surgery, they can be a source of morphological disorders, pain, and long-term adjacent degeneration [28,30].

The search for a good sagittal balance after AIS surgery therefore leads to a reflection on the target values of sagittal curvatures, on their planning, and on the intraoperative execution [31,32,33].

The most common way to bend rods is manually, without a measured target, based on the aim and experience of the surgeon [34,35]. The use of patient-specific rods (PSR) is undoubtedly a possible response to improve sagittal balance restoration and to obtain post-operative sagittal angles that are closer to the planned ones [36,37,38]. This method was first utilized for adults, then, more recently, for adolescents [31,39].

This review aims to provide an update about patient-specific planning and rods in AIS, looking at the current literature.

The following topics in the publications will be analyzed:-TK planning method;-Manufacturing: various ways to obtain PSR;-Comparison between programmed and achieved TK;-PJK incidence.

## 2. Materials and Methods

This is a systematic review of the patient-specific planning and rods for AIS surgical correction. It has been submitted and registered to the PROSPERO website https://www.crd.york.ac.uk/PROSPERO/ (accessed on 11 January 2024) with number 414039. This report was prepared according to Preferred Reporting Items for Systematic Reviews and Meta-Analyses (PRISMA) guideline 5, as suggested by the Enhancing the QUAlity and Transparency Of health Research (EQUATOR) network (Appendix A) [40].

As a literature review, ethics committee approval was not required.

Electronic databases of EMBASE, MEDLINE/PubMed, Science Direct, Scopus, and Web of Knowledge were searched from 2013 (first use of PSR) through 30 November 2023 (search date), with the following keywords: “adolescent” or “children” + “scoliosis” + “patient-specific” or “patient” and “specific”, including but not limited to reports in English, French, Italian, Spanish, and Portuguese languages. We also searched Google and Google scholar to bring out the gray literature, and further reviewed them for credibility after the initial search. The literature was further checked on 4 January 2024 lest miss a more recent paper.

For the systematic review, we aimed to analyze all case series and case reports of PSR and/or patient-specific planning for AIS, including journal articles and meeting proceedings. No minimum follow-up was defined for inclusion.

Articles were further screened for interventions and were included if they clearly reported the type of treatment and the radiologic sagittal outcomes at the last follow-up.

To be comprehensive, bibliographies of relevant reviews and selected studies were examined. Reviews, historical articles, and other related documents were manually added.

Study selection was performed in two stages by paired reviewers (first and last author), screening independently and in duplicate. Titles and abstracts were screened in the first stage, followed by full-text readings of potentially eligible citations.

The same-paired reviewers extracted the data independently and in duplicate using electronic data extraction forms. Disagreements were resolved by consensus or through discussion with a third investigator (second author). The selection of the articles is summarized in the PRISMA diagram (Figure 1).

The potential bias was assessed using the MINORS score, which evaluates non-randomized comparative studies with 12 questions and non-comparative studies with 8 questions, scoring them from 0 to 2 [41]. The sum of the points was used to grade the quality of each study: poor (<8 for non-comparative or <12 for comparative), good (9–12 for non-comparative or 13–18 for comparative), or excellent (>13 for non-comparative or >18 for comparative).

## 3. Results

### 3.1. Literature Search Results

Eight case studies of PSR and AIS were identified, exclusively from France (n = 5) or the United States (n = 3). One presented redundant data and deserved exclusion. Thus, seven studies were retained for the present analysis: four journal articles and three proceedings from international congresses, reporting on 355 patients in total [39,42,43,44,45,46,47]. An overview is provided in Table 1. The study design was mostly a cohort study (evidence level: 4), except for one comparative study versus standard rods (evidence level: 3) [48]. Six were monocentric and one was multicentric. The minimum follow-up was between 4 and 24 months.

The average MINORS score was 11.7 ± 2.2 (min. 9, max 15), resulting in good quality for six studies and excellent quality for one study (Table 1).

### 3.2. Radiological Planning and Analysis

Each author deliberately chose their target TK based on their experience, within ranges of 25–45°, with the highest values for the highest PI, but there was not a clear method of calculation [42,43,44,45,46,47]. The analyzed studies did not clearly report on TKA planning and the achieved position, nor on the number of vertebrae of TK and the transition points. The limits of TK measurements were T4-T12, T5-T12, global, i.e., maximum TK, or “instrumented” TK, i.e., kyphosis of the instrumented thoracic spine patients [42,43,44,45,46,47].

The programs used were mainly Surgimap^®^ and Unid Hub^®^ [49,50] (Table 2).

### 3.3. How to Obtain PSR

The analyzed literature showed that various strategies are currently available to obtain PSR, with each PSR company proposing its own spine fixation system (Table 3).

The most basic way to implant rods that are similar to the planned rods should be to contour them with a manual bender according to pre-operative planning. However, this process is potentially imprecise and few articles reported on it [34,51].

A slightly more precise option to obtain quasi-PSR is to choose “best match” rods from a set of pre-bent rods of various curves and lengths (Robert-Reid Inc., Tokyo, Japan) according to preoperative planning [52]. Sudo et al. did not report rod-cutting or additional bending [53,54]. However, if required, such modifications are possible in order to fit the length of the instrumented spine and the targeted alignment. This process should be more precise than manual bending since the rods’ curve is industrially measured, but the plan would probably match some approximation of the shape of pre-bent rods. Moreover, the plan would be more expensive than manual bending but there would be no notches, other than in cases of additional manual bending. Data from the literature about this process showed good sagittal results but did not address the relationship between the planning and the achieved sagittal alignment [52,53,54]. In a comparative study, patients with notch-free pre-bent rods had a significantly higher postoperative TK than patients with conventional, manually bent notched rods (30 vs. 24°). The rod deformation angles were significantly lower in the notch-free rods than in the notched rods on the concave side (7 vs. 13°) [52]. These results suggest that the notch-free rod can better maintain its curvature, leading to the better correction or maintenance of TK than the notched rod. To the best of our knowledge, this type of implant is only available in Japan [55].

A third option to obtain patient-specific contouring is to print a paper template in 1:1 dimensions using the digitally planned rod [42,43]. This can then be used in the operating room in a sterile envelope, allowing for the surgeon to bend the rods accordingly. Two articles are available on this process, showing a post-operative TK within +/− 5.5° of the predicted value from Marya and no significant difference for Ferrero [42,43]. Marya also reported an average under-bending of rods of < 1°. Ferrero reported good correspondence between planned and achieved lordosis (57~58°), with constructs reaching L2, L3, or L4, but different TL inflection points of about two levels between planned and implanted rods [42,43]. This process of obtaining PSR presents no additional cost compared with standard rods; however, it requires more time during surgery than pre-bent rods, and notching will be present.

Another available option to obtain the measured rod contouring during the surgery is to use a calibrated bender linked to a planning program (Bendini, Nuvasive^®^, San Diego, CA, USA) [56]. This process is probably precise, and is somewhat expensive due to the connected bender, but is potentially less precise and less expensive than factory-bent rods. However, notches will be present and there are currently no available results on this system’s use in AIS surgery.

Finally, the most sophisticated and, probably, most precise system is the industrial manufacturing of pre-bent rods according to planning. The first company to develop this process was Medicrea, a French company of spine implants including side-connected polyaxial dome screws. Five reports are available on this [39,44,45,46,47]. In 2021, Medtronic, an international company of medical technologies including spine implants, acquired Medicrea, and currently proposes PSR’s use for the side-connection Medicrea system (PASS LP^®^) and for the Medtronic top-connection tulip screw and hook systems (Solera^®^) (Table 3) [50]. In 2021, SMAIO, another French company of spine implants and programs, developed its own PSR manufacturing system, with side-connected monoaxial dome screws [57]. For these implants, there are currently no available results. This process should be the simplest for the surgeon and is probably the most precise, with no notching on the rods; however, it is quite expensive, since it requires specific manufacturing for each patient (as per “haute couture” clothing).

### 3.4. Radiological Outcomes

From the analyzed studies, coronal correction was between 64% and 75% [8,42].

The rate of patients with postoperative normokyphosis was between 95% and 100% [42,43,44,45,46,47] (Table 4). Solla et al. reported that factors associated with achieved TK at the last follow-up included the concave rod contouring angle and the pre-operative TK angle (*p* < 0.05) [46]. The mean difference between the pre-operative TK and the TK at last follow-up was between −1° for the Cantilever technique [43] and 14° for postero-medial translation [46]. In hypokyphotic patients, the mean difference between the pre-operative and the last follow-up TK was between 14° for Cantilever technique and 20° for PMT [43,46].

Three studies reported no significant difference between the planned and achieved TK using sublaminar bands at the apex of the thoracic curves [42,44,45]. The behavior of hooks or claws at the apex of the main curve is not described with PSR. However, their use at the cranial part of the thoracic construct is reported in four studies [42,44,46,48].

The mean gap between planned and achieved TK was −3° for Thomas with 6 mm Ti rods, 0° for Solla (*p* = 0.85) with 6 mm CrCo rods, and Abelin with 5.5 CrCo or 6 mm Ti rods, 1° for Ferrero (*p* = 0.98) with 5.5 mm CrCo rods, and 5° for Marya (*p* = 0.4) with 5.5 mm Ti rods [42,43,44,45,46,47,48]. These data moderately suggest that using stiffer rods increases the correspondence between the PSR contour and the achieved TK.

Concerning the rods’ behavior, Thomas et al., using sublaminar bands and 6 mm Ti symmetrical PSR, reported a minimal change (<1 mm), even in the hypokyphotic group, in rod deflection at 2-year follow-up, compared to the predicted rod deflection [44]. From Alijanipour et al., both maximal deflection distance (23 vs. 17 mm) and the angles of tangents to rod endpoints (30 vs. 17°) were higher for PSR than for conventional rods [39]. Solla et al. reported a visual flattening of the concave rod but did not report specific measurements and suggested the concave rods were over-contoured by 10° in cases of pre-operative hypokyphosis [46]. Concerning subgroup analysis, from over-bent 6 mm CrCo concave side rods, the mean TK gain was 20° for an expected gain of 25° in the subgroup with pre-operative hypokyphosis (<20°). Of the 17 patients in this subgroup, 10 were under-corrected (achieved TK 5° lower than expected TK) but all achieved TK > 20°. However, in the subgroup with normal preoperative kyphosis (n = 18), the mean TK gain was 8° for an expected gain of 4°. In this subgroup, 11 out of 18 were overcorrected (achieved TK was 5° higher than expected TK) [46].

In a study with 5.5 Ti rods and multiple screws, there was a significant post-operative change in TK in both the hypo- and hyper-kyphotic patient groups, resulting in patients achieving a mean TK within the ‘normal’ parameters of 20–40°, whereas the normokyphotic patients had a marginal, non-relevant increase in TK post-operatively [43].

The thoraco–lumbar junction was specifically analyzed in three studies: two of them obtained a straight TL junction after PSR surgery, whereas Thomas found an average of 8° of lordosis at the last follow-up, very close to the pre-operative value (7°) [43,44,47]. They also found that the sagittal TL inflection point in hypokyphotic patients shifted inferiorly, from the T9 superior endplate preoperatively to the T10 superior endplate postoperatively, which was maintained throughout the 24-month follow-up. However, the planned position of TKA was not declared.

Concerning lumbo-pelvic parameters, Ferrero reported that 21% of the patients had not achieved LL within reference values: four had hypolordosis and six had hyperlordosis [42]. Nevertheless, in 25% (n = 12), the 3D planning tool overestimated lumbar lordosis by 10° or more. Postoperative SVA was superior by 20 mm in nine cases (19%) and the C7 plumb line was anterior to the sacrum in 33% of cases (n = 16). Nevertheless, the postoperative values of TK, pelvic tilt, and SVA were not different from the planned values. Thomas also reported an LL increase, which was mainly observed in L1–L4, with no significant change in L4–S1. The pelvic parameters remained relatively unchanged. These authors observed a pelvic retroversion (PT increase and SS decrease) at 6 months, which returned to baseline at 12 or 24 months post operation. Thomas et al. found a compensatory median gain of 7° in LL by the 2-year follow-up, reaching “normal” parameters as proposed by Mac-Thiong et al. [44,58].

According to the available data, there is no PJK after PSR implantation.

## 4. Discussion

All authors reported high correspondence between planned and achieved TK, despite the use of different surgical techniques and rod properties.

Concerning the planned TK, the previous literature [59,60] suggested that achieving ≥23° or ≥26° of TK decreased the risk of sagittal plane decompensation and cervical malalignment following thoracic fusions for AIS. However, the best target TK for each patient was rarely explored. It seems difficult to deduce the “ideal” sagittal alignment of the spine from the deformed spine sagittal alignment, and only the pelvic parameters can provide proper orientation [3]. Abelin-Genevois et al. found that the restoration of lumbo-pelvic alignment helped to limit early degenerative changes in the free motion segments after AIS surgery [61]. This systematic review has also confirmed that the pelvic parameters are not modified by surgery at follow-up, despite some transient post-operative changes [62]. It is therefore possible to predict the best spinopelvic alignment from pre-operative pelvic parameters, as in adult spines [63,64].

Conversely, both cervical and lumbar lordosis are negatively affected by pathological thoracic kyphosis [62,65]. Postoperative TK increases have been shown to achieve a reciprocal increase in LL that has beneficial effects for a patient’s future, related to the natural loss of LL with aging and disc degeneration [21]. Thomas et al. found a compensatory median gain of 7° in LL, reaching “normal” parameters [44,58]. From a previous study, Clement et al. reported an LL gain that is equal to approximately 40% of TK gain, with all the gain in proximal lumbar lordosis (PLL), while distal lumbar lordosis (DLL) equivalent to SS remained unchanged from preoperative measurements [62]. Conversely, the postoperative loss of TK is strongly associated with the reciprocal loss of LL [66]. In the same way, the increase in TK entailed an improvement in cervical lordosis related to the increase in distal cervical lordosis, with 60% of the TK increase transferred to the gain in distal cervical lordosis [65].

It has been geometrically demonstrated that global LL and GTK are dependent on pelvic parameters. The formula GTK = 2×(PT+LL-PI) has been validated in adolescents and young adults without spine pathology [24,25]. Therefore, each individual has a specific TK according to their lumbo-pelvic parameters. At present, it seems necessary not to choose a given target angle for all patients (e.g., 30°), but rather to seek the correct sagittal alignment by providing the patient’s “best” GTK. The calculation of the targeted GTK requires anticipating the post-operative variations in LL due to the increase in TK [62]. Then, it is easy to calculate the value of the instrumented TK from a targeted GTK.

The analyzed studies correspond to the beginning of PSR use, when the formula GTK = 2×(PT-LL-PI) was unknown. Each author deliberately chose a target TK based on their experience, within ranges of 25–45°, with the highest values for the highest PI, but there was not a clear method of calculation [42,43,44,45].

When planning for LL, a similar process is available. LL can be divided into PLL and DLL [62]. PLL is calculated using the formula PT+LL-PI, considering the increase in LL linked to the increase in TK.

The length of the TK and the position of the apex should also be planned. A TKA position between T5 and T8 is frequent in the normal population. A recent study suggests apex on T8 for mild PI and T9 for high PI (type 3 or 4 of Roussouly) [25]. Other authors suggest apex on T7 or T9 [58,67]. However, the ideal TKA position for each subject is still unknown, and depends on the length of GTK between its two points of inflection, and on the harmony of the kyphosis. If the kyphosis is regular, PTK is similar to DTK, and the apex is in the middle of GTK. On the other hand, if the kyphosis is not regular, PTK and DTK are not equal, and the apex is shifted up or down. Unfortunately, the analyzed studies did not clearly report on TKA planning and the achieved position, nor did they report on the number of vertebrae of TK and the transition points. We recommend a more complete and specific assessment of sagittal results with an evaluation of the type of Roussouly, the apex of the sagittal curves, and the points of inflection.

Various options are currently available to plan sagittal correction and to implant rods corresponding to planning, ranging from the simplest and cheapest (printed rod model and manual bending) to the most expensive and precise (industrial manufacturing).

To improve the planning process, simulation tools allow for a clear definition of a targeted alignment for each patient (Table 2) [57,68,69]. Based on the literature, they seem useful for planning sagittal correction and anticipating the postoperative behavior of the corrected spine, regardless of whether the spine surgeon uses PSR [70]. Various programs are available for this purpose, with each having pros and cons: some are “independent”, whereas others are linked to a specific company producing implants. Based on the principle of balance between the pelvis, LL, and TK, it is possible to simulate the GTK correction conforming to a balanced sagittal alignment.

TK planning requires a clear definition of the measurement limits, which vary across the literature, e.g., T4–T12, T5–T12, T2–T12, T1–T12, global, i.e., maximum TK, and “instrumented” TK, i.e., kyphosis of the instrumented thoracic spine patients [2,8,20,22]. This variety of measurements is somewhat confusing, even for experienced readers. From the surgical point of view, the most objective and reliable parameter is probably the “instrumented” TK, i.e., the TK of the instrumented thoracic spine, which strictly reflects the adherence (or lack of) between the planned instrumented TK and the TK achieved for the instrumented zone. However, this measure is rarely reported [46]. From the functional point of view, the most comprehensive way to assess a patient’s alignment is certainly global thoracic kyphosis (GTK), which is the spinal segment in kyphosis that intervenes in the sagittal balance and is measured from the cervico-thoracic inflection point to the thoraco-lumbar inflection point [8,22,24]. GTK is characterized as having the most cranial and most caudal vertebrae, and by the position of the thoracic kyphosis apex (TKA). The horizontal line through the TKA separates GTK into proximal TK (PTK) and distal TK (DTK). However, after fusion, GTK may include both instrumented and uninstrumented thoracic segments, unless the construct covers the entire thoracic spine. Furthermore, in the case of PJK, GTK includes both instrumented TK and PJK. This concept highlights the need to measure proximal junctional angle (PJA, i.e., the sagittal angle between the proximal endplate of the upper instrumented vertebra and the superior endplate of the two supra-adjacent vertebrae above it) when assessing post-operative sagittal outcomes [26,28].

It must be pointed out that patient-specific planning requires more time than a lack of planning, for the PSR company and/or for the surgeon, but various articles and common sense suggest that planned surgery provides better outcomes than unplanned surgery [71,72]. Additionally, the planning should be prepared before the surgical procedure, allowing for the surgeon to concentrate on planning when outside the operating room and on the patient once inside [73,74].

Concerning surgical use, PSR can be implanted and connected to spine anchors like normal rods. However, the surgical strategies, the release technique (facet resection, osteotomies), the baseline characteristics, and a surgeon’s skills and experience could influence the relationship between the shape of the rod and the achieved sagittal alignment [75,76,77].

Monoaxial screws, if implanted parallel to the superior plateau, should pull each vertebra perpendicular to the rod and achieve a good spine adherence to rod shape; however, this comes at the cost of bending stress, and is potentially detrimental to the stability of the screws [78]. Contrarily, monoaxial screws with a “quirky” direction not parallel to the endplate should increase the work required to connect them to the rod and entail a less precise adherence to the planned alignment. These statements are less absolute for polyaxial screws, which tolerate a certain amount of obliquity and are less constraining but should result in a less precise congruence with the planned alignment [79]. However, there are currently no reports on PSR and monoaxial screws.

Moreover, the type of connection between screws and rods (top-loading vs. lateral connection) may influence the relationship between the shape of the rod and the achieved sagittal alignment, with side-connections probably providing better congruence in the case of severe sagittal disorder [80].

Three studies reported encouraging outcomes regarding the use of sublaminar bands at the apex of thoracic curves. Thomas et al. postulated that the use of sublaminar double bands in the area of apical hypokyphosis associated with postero-medial translation (PMT) resulted in a minimal change in rod shape. Similarly, both Grobost and Ferrero reported no difference between the simulated model and the postoperative sagittal parameters [42,44,45].

In hypokyphotic patients, the mean difference between pre-operative TK and TK at last follow-up was higher with the translation technique than for the cantilever. It is worth noting that both the minimum and maximum values of the difference between expected and achieved TK concerned screw-based constructs, suggesting that the type of vertebral implants is less relevant than the aim of the surgeon and the correction technique [43,46]. In the previous literature, the correction technique seems to play an important role. A recent multicenter study on 562 AIS showed that in situ bending and cantilever resulted in a postoperative decrease in TK of about 5°, whereas rod rotation and PMT resulted in an increase in TK (of +7° and +16°, respectively) [8]. It therefore seems better to use a reduction technique capable of reaching a target TK, especially in the case of hypokyphosis. Moreover, six out of seven studies from the present review concern the PMT technique, which seems more effective in adapting the spine to the plan and not the plan to the existing sagittal disorder [41,43,44,45,46,47]. On the other hand, in situ bending should not be used as the main correction technique with PSR because it implies per-operative rod contouring. However, a certain amount of in situ bending could be added after PMT or rod rotation to increase coronal correction, but this potentially decreases sagittal correction [8]. Conversely, if the surgeon wants to continue using their preferred cantilever technique, we would suggest over-bent rods, especially in cases of hypokyphosis [8].

Rod-flattening was frequently observed due to a compromise between the stiffness of the spine and the corrective power of the construct, especially in severe pre-operative hypokyphosis [46]. This can be anticipated, at least for moderate AIS and reproducible surgical techniques. With conventional rods, Cidambi et al. reported rod-flattening with a decrease in deflection of 13 mm and a 21° decrease in rod angle with 5.5 mm stainless steel rods [81]. Abe et al. reported a rod-flattening of 16° in patients treated with 6 mm Ti rods [82]. Kluck et al. reported that concave rods flattened, on average, by ~20°, whereas the average convex rod angle increased by 4° [83]. Sia et al. reported that the curvature of the titanium rod and cobalt chrome rod decreased from 60° to 37°, and 51° to 28°, respectively [84]. Le Naveaux and Gay recommended over-contouring the concave rod by 13° to induce an increase in postoperative TK and apical derotation [85,86].

In the available data, there is no PJK after PSR implantation. Even if this complication is underreported, it has been specifically assessed in three studies [41,42,43], whereas the common rate from the literature is between 7% and 46% [13,28,29,30]. Despite the multifactorial etiology, a good sagittal alignment is confirmed to be a strong protective factor [29,87]. The use of hooks or claws at the proximal part of the thoracic construct could have played a role in the absence of PJK, as previously reported with standard rods [88,89,90].

## 5. Limitations

The limits of the current review include the small number of subjects in the published studies. Moreover, most papers suffer from industry support, a moderate level of evidence (3 or 4), the short and different lengths of the observation periods, a moderate risk of bias with only one comparative study, and the limited amount of available data. Furthermore, the TK measurements are not the same for all studies.

## 6. Future Directions

The next steps should include multicenter studies using various surgical and manufacturing strategies to assess:-How the properties of the rod (diameter, section, material, notched vs. not notched), surgical factors (type and density of implants, type of rod–screw connection, correction and release technique), and baseline variates (spine stiffness, pre-operative TK, patient-related factors, etc.) might influence the relationship between the plan and the achieved alignment;-If the achieved plan, including the regularity of TK, the position of the apex, and the transition points between TK and adjacent curves, was optimal concerning the postoperative modifications to global alignment, adjacent sagittal curves, and quality of life. For this, TK planning requires a clear definition of the measurement limits, apex, and the number of vertebrae included.

To fine-tune planning, sagittal results should be predictable at both instrumented and uninstrumented levels. If the latter are known from the literature, the former should be analyzed for each surgeon, depending on the implants, the correction technique, and human factors.

Further clinical evaluations are underway to confirm the benefits of planning sagittal results and implanting PSRs that are strictly bent following the planning, allowing for a quantifiable and reproducible sagittal correction.

## 7. Conclusions

Various options are currently available to plan sagittal corrections and to implant rods corresponding to planning.

The outcomes of the first PSR experiences in AIS surgery are encouraging, showing a good correspondence between the expected TK and the achieved TK, and the absence of PJK.

Current data suggest using stiff, over-bent, concave side rods, and translation techniques for correction, in cases of preoperative hypokyphosis.

## Figures and Tables

**Figure 1 children-11-00106-f001:**
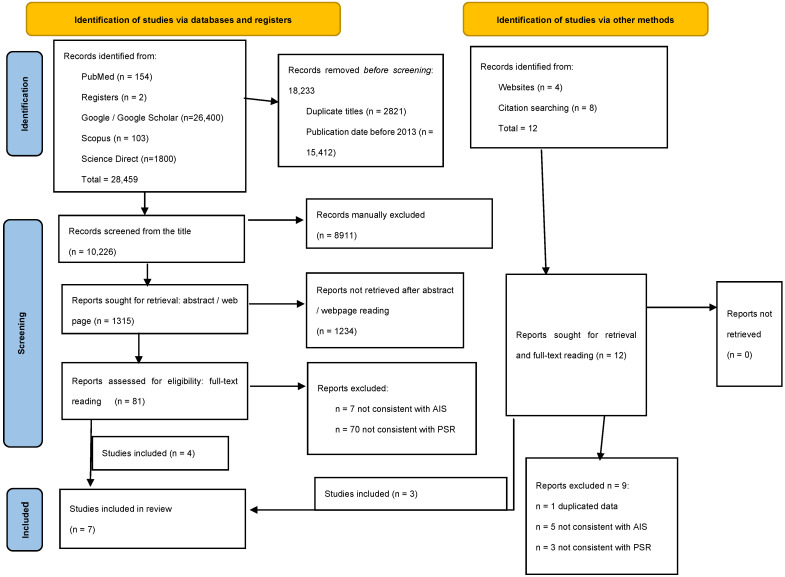
PRISMA flowchart.

**Table 1 children-11-00106-t001:** Quality assessment.

General Informations				MINORS Sub-Score										
Main Author	Type of Study *	Minimum Follow-Up (Months)	Type of Paper **	A Clearly Stated Aim	Inclusion of Consecutive Patients	Prospective Collection of Data	Appropriate Endpoints	Unbiased Assessment of the Endpoint	Follow-Up Period	Loss to Follow Up < 5%	Adequate Statistical Analyses	Total	Out of	Quality
Thomas	1	24	A	2	2	1	2	0	2	1	2	12	16	Good
Marya	1	6	A	2	2	1	2	1	1	0	2	11	16	Good
Solla (OTSR)	1	12	A	2	2	2	2	1	1	1	2	13	16	Excellent
Solla (ESJ)	2	12	P	2	0	1	2	1	1	0	2	9	16	Good
Alijanipour	1	12	P	2	2	1	2	0	1	0	2	15 ***	24	Good
Grobost/Abelin	1	6	P	1	2	2	2	1	1	0	1	10	16	Good
Ferrero	1	4	A	2	2	2	2	2	0	0	2	12	16	Good
Average		10.86		1.86	1.71	1.43	2.00	0.86	1.00	0.29	1.86	11.17	16	
Modal value	1	12	A	2	2	1	2	1	1	0	2	12		Good

* 1 = monocentric, 2 = multicentric. ** A = Article, P = Proceeding. *** 10 points from listed items + 5 points for comparative study with adequate control group with same baseline characteristics.

**Table 2 children-11-00106-t002:** Alignment planning programs.

Program	Online/Downolad?	Free/Suscription	Owner	Link to Spine Companies	Planning Author	Pros	Cons
Surgimap	Download	Basic version is free	Independent	Stryker, Globus, various	Surgeon	Free version	
Keops	Online	Subscription (but usually free for SMAIO clients)	Smaio	Smaio	SMAIO Company	Possible data sharing for scientifc studies; Radiological analysis by a third part	
Unid hub	Online	Free for Medtronic clients	Medtronic	Medtronic only	Surgeon and/or Medtronic team	Radiological analysis by a third part	Hard password; only for medtronic planning
SpineEOS	Online	Subscription	Alphatec	None	Surgeon	Link to EOS imaging	Need for EOS imaging

**Table 3 children-11-00106-t003:** Companies involved in PSR and/or pre-bent rods.

Company (Country)	Type of Technology	Type of Rods	Rod–Screw Connection	Fixation Implants
Medicrea (Fr)/Medtronic (US)	Planning and manufacture	Ti or CoCr, 6 or 5.5 or 3.5 mm, round or derotation rod with baseball-field section (2 plate faces and 2/3 of circus)	-Top connection (tulip screws)	Polyaxial, monoaxial or uniplanar pedicle screws;
			-Side connection (dome screws) with polyaxial/derotation/realignment connectors	Hooks, claws, sublaminar bands
SMAIO (Fr)	Planning and manufacture	Ti 6 or 5.5 mm, round section	Side connection	Monoaxial;
				screws, hooks and claws.
Nuvasive (US)	Planning and measured bending with a connected bender	Ti or CrCo, 6 or 5 mm, round section	Top connection	Polyaxial or monoaxial; screws, hooks, sublaminar bands
Robert Reid (Japan)	Manufacturing of pre-bent rods	CrCo 5.5, round section	Side connection	Polyaxial screws

Fr: France; Ti: Titanium; CrCo: Chromium–Cobalt.

**Table 4 children-11-00106-t004:** Overview of the analyzed studies.

Main Author	Year	Planning Software	Rods Material and Technology	Pre-Bent or Manually Bent	Surgical Technique and Construct	Number of Patients	Coronal Cobb Angle	TK Increase	TK Increase in Hypo-TK	Planned TK	Planned–Achieved TK	% Patients with Normal TK at Last Follow-Up	Postoperative TL Angle
Thomas [44]	2022	Unid hub	6 mm Ti, identical, Unid	Pre-bent	ST2R with Ponte osteotomies, apical sublaminar bands (n = 4)	48	63	6.4	19	30 to 40°	−3°		8° lordosis
Marya [43]	2023	Surgimap	5.5 mm Ti, asymetrical (+20° on concave side), manually bent according to a paper template; rail on concave side, round on convex	Manually bent	Cantilever, multiple pedicle screws construct	61	68	−1	14		5° ± 4		
Solla [46]	2018	Surgimap	6 mm CoCr, asymetrical, diamond section, Unid (+10° for concave side rod)	Pre-bent	ST2R, multiple pedicle screws construct, concave derotation	37	53	14	20	34	0°: −4 in normoK, +5 in hypo K	97% (1 patient with TK = 56°)	
Solla [47]	2020	Surgimap or Unid Hub	Unid, various: 5.5 or 6 mm, Ti or CoCr	Pre-bent	ST2R, multiple pedicle screws construct ± concave de-rotation or sublaminar bands	85	-	12	19		1°: −4 in normoK, +6 in hypo K	96% (2 patients with TK between 10 and 20°)	
Alijanipour [39]	2017	Surgimap	Mostly 6 mm Ti, identical, Unid vs. conventional “unplanned” rods	Pre-bent	ST2R with multiple pedicle screws construct	28 vs. 28	57	−2 vs. −3					significantly lordotic in C group (−7.3_) compared to PS group (−0.3_, p\0.001).
Grobost/ Abelin [45]	2019	Keops	5.5 mm CoCr or 6 mm Ti, identical, Unid	Pre-bent	ST2R with multiple pedicle screws construct + sublaminar bands at the apex	49	54 ± 10	10		30 ± 8	0	95%	significantly improved after surgery
Ferrero/ Ilharreborde [42]	2018	SpineEOS	5.5 mm CoCr identical manually bent according to a paper template	Manually bent	Translation on 1 rod; lumbar pedicle screws and thoracic sublaminar bands	47	59 ± 13	9		38	1	100%	
Sum						355							
Average	2020		5.8			51	59	7	18	34	0.67	97	
Modal value		Surgimap		Pre-bent	ST2R								

ST2R: simultaneous translation on two rods; TK: thoracic kyphosis.

## Data Availability

All data are included in the article or in the tables.

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
