# Peer review of "Patient-Specific Surgical Correction of Adolescent Idiopathic Scoliosis: A Systematic Review"

_children, 2024, doi:10.3390/children11010106_

Round 1
Reviewer 1 Report
Comments and Suggestions for Authors
Dear Author,
the manuscript "Patient-Specific Surgical Correction of Adolescent Idiopathic Scoliosis: an Update" is interesting and presents an important topic.
General information:
-first of all, prepare the article for review in accordance with the journal's guidelines, especially references.
-in the key words section, please do not repeat words from the title.
Introduction:
-in first paragraph: references should be precisely matched to the information in this paragraph, not [1-7]. Please check it. It should be corrected.
-in line 48/49: "In addition to coronal outcomes, sagittal results strongly affect long-term quality of life and degeneration of adjacent levels [9-17]." In my opinion there are too many references for this sentence. It should be corrected.
-in line 53/54: "Moreover, many publications reporting AIS postoperative outcomes have emphasized the risk of thoracic hypokyphosis after posterior fusion [8-18]." Does this sentence require that many references? Please check it.
-please check the aim of the study, it is very complicated and difficult to interpret for a normal reader. It should be corrected.
Literature search results
-Table no. 2 should be no. 1, it should be corrected.
-Additionally, please enter the reference number of the cited studies in this table 2 (1).
-Please correct the numbering of the following tables.
Surgical strategy
-in line 290/293 "PSR can be implanted and connected to spine anchors like normal rods. However, surgical strategies, release technique (facet resection, osteotomies), baseline characteristics, and a surgeon’s skills and experience could influence the correspondence between the shape of the rod and the achieved sagittal alignment [69, 73-80]."
Do this sentences require that many references? Please check it.
Radiological outcomes
-in line 331/332 "From the analyzed studies, coronal correction was similar to that obtained with conventional rods (64 to 75%) [42-48]."
Does this sentence require that many references? Please check it.
Limitations
-In limitations, it is worth mentioning the different lengths of observation periods.
Conclusions
Conclusions sounds like a discussion, I think it should be improved. Conclusions should be short and clear to the reader.
There are too many citations in the manuscript. In my opinion this should be corrected first. The article requires the additional changes listed above.
Author Response
Dear Author,
the manuscript "Patient-Specific Surgical Correction of Adolescent Idiopathic Scoliosis: an Update" is interesting and presents an important topic.
Reply: Thank you for your comment
General information:
-first of all, prepare the article for review in accordance with the journal's guidelines, especially references.
Reply: OK, I will.
-in the key words section, please do not repeat words from the title.
Reply: The keywords section has been modified as requested. Thank you for this suggestion.
-There are too many citations in the manuscript. In my opinion this should be corrected first. In first paragraph: references should be precisely matched to the information in this paragraph, not [1-7]. Please check it. It should be corrected.
In line 48/49: "In addition to coronal outcomes, sagittal results strongly affect long-term quality of life and degeneration of adjacent levels [9-17]." In my opinion there are too many references for this sentence. It should be corrected.
In line 53/54: "Moreover, many publications reporting AIS postoperative outcomes have emphasized the risk of thoracic hypokyphosis after posterior fusion [8-18]." Does this sentence require that many references? Please check it.
In line 290/293 "PSR can be implanted and connected to spine anchors like normal rods. However, surgical strategies, release technique (facet resection, osteotomies), baseline characteristics, and a surgeon’s skills and experience could influence the correspondence between the shape of the rod and the achieved sagittal alignment [69, 73-80]." Do this sentences require that many references? Please check it.
In line 331/332 "From the analyzed studies, coronal correction was similar to that obtained with conventional rods (64 to 75%) [42-48]." Does this sentence require that many references? Please check it.
Reply: we reduced the references where suggested and precisely matched. However, the instructions to authors suggest >80 references, and the editorial board has even suggested to increase them.
-please check the aim of the study, it is very complicated and difficult to interpret for a normal reader. It should be corrected.
Reply: The aim section has been modified as requested
Literature search results
-Table no. 2 should be no. 1, it should be corrected. -Additionally, please enter the reference number of the cited studies in this table 2 (1). Please correct the numbering of the following tables.
Reply: We are sorry for this. The tables have been modified as requested
Surgical strategy
-In limitations, it is worth mentioning the different lengths of observation periods.
Reply: Thank you for this comment. The different lengths of observation periods has been added as a limit.
Conclusions sounds like a discussion, I think it should be improved. Conclusions should be short and clear to the reader.
Reply: Thank you for this comment. In the first version, there was no “discussion” section. In the new version, the conclusion is shorter and clearer.
Reviewer 2 Report
Comments and Suggestions for Authors
Dear authors,
In this systematic review, you sought to systematically review patient specific rods for surgical correction of AIS.
General comments
First of all, there are minor language mistakes that need addressing. Also, the abstract of the study lacks adherence to Systematic Review guidelines. Another issue is the structural organization of the paper. As per the Journal’s instructions, you should be presenting a Results and a Discussion section. In the discussion section, I would advise you to not report statistical results from the individual papers. Other than that, the article reads well and the scientific quality is satisfactory from an orthopaedic point of view.
Please find some more specific comments below.
Title: The title needs to include the study design. Consider revising.
Abstract: First of all, the level of evidence should be better defined as 4 (ie the lowest level of evidence of the included papers determines the SR level of evidence). Also, the literature search date needs to be reported in the abstract. Presentation of implications for future research should be avoided on the abstract. Also, you will need to define whether the cohort studies were prospective or retrospective in nature.
Comment on PRISMA 2020 flow chart
Could you explain why a Google search is a reliable method to identify relevant studies?
Classeur1 biblio results supplemental file is poorly presented, and I would advise you do your own table by using the appropriate tools. Also you have not spelled out the table abbreviations.
Limitations section
Please revise this paragraph and elaborate on the shortcomings. You will need to divide the long sentence into shorter ones for clarity.
Comments on the Quality of English LanguageMinor polishing is required.
Author Response
Dear authors,
In this systematic review, you sought to systematically review patient specific rods for surgical correction of AIS.
General comments
First of all, there are minor language mistakes that need addressing.
Reply: A native Anglophone has edited the text
Also, the abstract of the study lacks adherence to Systematic Review guidelines.
Reply: Thank you for this comment. The abstract of the study has been rewritten according to Systematic Review guidelines.
Another issue is the structural organization of the paper. As per the Journal’s instructions, you should be presenting a Results and a Discussion section. In the discussion section, I would advise you to not report statistical results from the individual papers.
Reply: Thank you for this comment. In the first version, results and comments were presented together. In the new version, there is a “results” section with data and a “discussion” section with comments, insights, and comparison with previous literature.
Other than that, the article reads well and the scientific quality is satisfactory from an orthopaedic point of view.
Reply: Thank you for your review
Please find some more specific comments below.
Title: The title needs to include the study design. Consider revising.
Reply: Thank you for this comment. The title has been modified as follows: Patient-Specific Surgical Correction of Adolescent Idiopathic Scoliosis: a Systematic Review.
Abstract: First of all, the level of evidence should be better defined as 4 (ie the lowest level of evidence of the included papers determines the SR level of evidence).
Reply: the level of evidence is defined as 4 as suggested
Also, the literature search date needs to be reported in the abstract.
Reply: we have added the date and period.
Presentation of implications for future research should be avoided on the abstract.
Reply: we removed implications for future research
Also, you will need to define whether the cohort studies were prospective or retrospective in nature.
Reply: we have added the nature of the studies
Comment on PRISMA 2020 flow chart
Could you explain why a Google search is a reliable method to identify relevant studies?
Reply: we used it to explore grey literature and search for proceedings. We added this info in methods section.
Classeur1 biblio results supplemental file is poorly presented, and I would advise you do your own table by using the appropriate tools. Also you have not spelled out the table abbreviations.
Reply: thank you for this comment. We have modified as suggested.
Limitations section
Please revise this paragraph and elaborate on the shortcomings. You will need to divide the long sentence into shorter ones for clarity.
Reply: we revised as suggested.
Comments on the Quality of English Language: Minor polishing is required.
Reply: A native Anglophone has edited the text
Round 2
Reviewer 1 Report
Comments and Suggestions for Authors
Dear Authors,
thank you for the changes you made.
However you should respond to all the reviewer's points.
- "-first of all, prepare the article for review in accordance with the journal's guidelines, especially references." - this has not been corrected. Please prepare references in accordance with the journal's guidelines.
- "Reply: we reduced the references where suggested and precisely matched. However, the instructions to authors suggest >80 references, and the editorial board has even suggested to increase them."
Please understand the reviewer correctly. Authors can cite a lot of research. However, a specific citation needs to be matched to specific information, which has not been done before. Now, it looks much better.
Apart from that I have no further comments.
Best regards,
Reviewer
Reviewer 2 Report
Comments and Suggestions for Authors
Dear authors,
Thank you for submitting the revised version of your paper. Please note you have now addressed my comments.